# Flat Magnetic Stimulation for Stress Urinary Incontinence: A 3-Month Follow-Up Study

**DOI:** 10.3390/healthcare11121730

**Published:** 2023-06-13

**Authors:** Marta Barba, Alice Cola, Giorgia Rezzan, Clarissa Costa, Tomaso Melocchi, Desirèe De Vicari, Stefano Terzoni, Matteo Frigerio, Serena Maruccia

**Affiliations:** 1Department of Gynecology, IRCCS San Gerardo dei Tintori, University of Milano-Bicocca, 20900 Monza, Italy; alice.cola1@gmail.com (A.C.); g.rezzan@campus.unimib.it (G.R.); c.costa14@campus.unimib.it (C.C.); t.melocchi@campus.unimib.it (T.M.); d.devicari@campus.unimib.it (D.D.V.); frigerio86@gmail.com (M.F.); 2Department of Urology, ASST Santi Paolo e Carlo, San Paolo Hospital, 20142 Milano, Italy; stefano.terzoni@asst-santipaolocarlo.it (S.T.); serena.maruccia@gmail.com (S.M.)

**Keywords:** quality of life, stress urinary incontinence, magnetic stimulation, pelvic floor disorders

## Abstract

Background: flat magnetic stimulation is based on a stimulation produced by electromagnetic fields with a homogenous profile. Patients with stress urinary incontinence (SUI) can take advantage of this treatment. We aimed to evaluate medium-term subjective, objective, and quality-of-life outcomes in patients with stress urinary incontinence to evaluate possible maintenance schedules. Methods: a prospective evaluation through the administration of the International Consultation on Incontinence Questionnaire-Short Form (ICIQ-SF), the Incontinence Impact Questionnaire (IIQ7), and the Female Sexual Function Index (FSFI) was performed at three different time points: at the baseline (T0), at the end of treatment (T1), and at 3-month follow-up (T2). The stress test and the Patient Global Impression of Improvement questionnaire (PGI-I) defined objective and subjective outcomes, respectively. Results: 25 consecutive patients were enrolled. A statistically significant reduction in the IIQ7 and ICIQ-SF scores was noticed at T1 returned to levels comparable to the baseline at T2. However, objective improvement remained significant even at a 3-month follow-up. Moreover, the PGI-I scores at T1 and T2 were comparable, demonstrating stable subjective satisfaction. Conclusion: despite a certain persistence of the objective and subjective continence improvement, the urinary-related quality of life decreases and returns to baseline values three months after the end of flat magnetic stimulation. These findings indicate that a further cycle of treatment is probably indicated after 3 months since benefits are only partially maintained after this timespan.

## 1. Introduction

Pelvic floor disorders include a series of diseases associated with pelvic floor weakening, which involve bowel, urinary, supports, and sexual dysfunctions [1]. Obstetric trauma is considered the primary damage to the pelvic floor giving the predisposition to develop pelvic floor disorders [2]. However, changes in the composition and enzymatic activity in the connective tissue play a role in the genesis of pelvic floor disorders [3]. Some of these changes in the collagenic patterns have been related to the menopausal decrease in estrogen [4]. Since pelvic floor disorders share risk factors, specific conditions may coexist, recur, or evolve into others as a consequence of treatment, such as surgery [5,6]. For instance, overactive bladder symptoms tend to improve after prolapse repair but may worsen if a suburethral tape is positioned at the time of surgery [7].

Stress urinary incontinence (SUI) represents one of the most common and bothersome pelvic floor disorders. Almost 50% of women in developed countries are estimated to be affected, and the lifetime risk of undergoing surgery is about 4% [8,9]. SUI is characterized by involuntary leakage of urine when the intra-abdominal pressure increases more than the urethral closure pressure such as during coughing, effort, or sneezing [10]. Pathogenetic mechanisms involve injuries to the connective tissue of the urethra, leading to urethral hypermobility and intrinsic urethral deficiency [11]. In addition, stress urinary incontinence may also occur (or persist) after pelvic floor surgery [12,13]. Stress urinary incontinence negatively affects women’s quality of life in terms of social, domestic, and psychophysical well-being, with a negative effect on sexual function [14]. Urinary incontinence can reduce the opportunity to be part of intimate relationships, socialize, or the ability to perform daily activities [15]. SUI diagnosis and management need great expertise to approach the intimate sphere of patients who are unable to express themselves autonomously. During the visit, the gynecologist must be able to discuss any concerns and assess any problems related to the quality of life and sexual well-being [16].

Urodynamics may be useful to confirm the diagnosis since clinical and instrumental findings poorly agree in the evaluation of bladder dysfunction [17,18]. However, its diagnostic importance in the work-up of urinary incontinence is currently debated due to differences in performance and adopted definitions [19,20]. Stress urinary incontinence management involves both surgical and conservative treatments based on the patient’s will, comorbidities, and quality-of-life impairment. According to the guidelines, conservative measures are considered the first-line choice, while surgical treatment is usually considered after the failure of conservative management. Different surgical options can be proposed for the treatment of SUI, such as anterior compartment repairs, bladder neck suspensions, midurethral slings, and injections [21,22,23,24,25]. To date, midurethral slings are considered the first option because of their high efficacy rates [26]. Retropubic tapes were introduced in 1995 and became the gold standard for SUI treatment [27]. To reduce the complications associated with the blind passage of needles in the retropubic space, the transobturator approach was developed in 2001 [28]. Finally, single-incision slings (SISs) were introduced in 2006. Their novelties were the shorter tape length and the limited intracorporeal dissection, avoiding the passage of tape and trocars through the obturator foramen, adductor tendons, and skin [24]. However, all surgical procedures have pitfalls, including visceral injuries, chronic pelvic pain, de novo bladder voiding dysfunctions, and overactive bladder symptoms [29,30]. As a consequence, conservative strategies should be preferred when possible. Options are represented by lifestyle modifications, pelvic floor exercises, electrical stimulations, biofeedback, and energy-based treatments [31].

An optional treatment for the treatment of stress urinary incontinence is represented by magnetic stimulation. Magnetic stimulators are extracorporeal devices that generate a specific electromagnetic field that interacts with pelvic floor neuromuscular tissue inducing intense muscular contractions and regulating neuromuscular control. Previous studies investigating magnetic stimulation for the treatment of female SUIs demonstrated a certain efficacy [32]. Specifically, systematic reviews and meta-analyses show significant improvements in quality-of-life questionnaires related to urinary incontinence [32,33]. In recent years, technological advancements have improved magnetic stimulator devices. One of them is represented by flat magnetic stimulation. This is characterized by homogeneous electromagnetic fields able to treat the entire pelvic area. In fact, this new magnetic field generates an equal distribution and intensity of stimulation. Consequently, flat magnetic stimulation allows for a large activation of muscle fibers without leaving areas of inconstant/suboptimal recruitment. This is thought to be associated with enhanced efficacy compared with standard magnetic stimulation treatment. The efficacy of this conservative treatment comes from the use of electromagnetic energy, the deep penetration of the waves, and the global stimulation of the pelvic floor. The magnetic field, through electrical tissue currents, induces changes in muscular contraction and allows neurons depolarization and blood supply enhancement. These modifications induce muscle fiber hypertrophy and hyperplasia due to more efficient stimulation. A previous experience has demonstrated the muscle hypertrophy of the urethral rhabdosphincter after flat magnetic stimulation, which has an established role in maintaining stress urinary continence. Similarly, preliminary reports of this new treatment option demonstrate exciting results in terms of quality-of-life improvements, but medium-term data, as well as optimal maintenance treatment schedules, are still unknown [34].

Consequently, the aim of our study is to analyze medium-term outcomes in patients with stress urinary incontinence undergoing flat magnetic stimulation in terms of objective and subjective cure rate and quality-of-life improvement and evaluate possible maintenance schedules.

## 2. Materials and Methods

This was a prospective interventional study. Recruitment occurred from August 2022 to September 2022 in the gynecologic outpatients at IRCCS San Gerardo dei Tintori Foundation in Monza, Italy. During the period of the study, a patient clinical interview to investigate the presence of lower urinary tract symptoms, such as urge urinary incontinence (UUI), stress urinary incontinence (SUI), overactive bladder (OAB), voiding symptoms (VS), or prolapse symptoms or anal incontinence was performed. All definitions conformed to IUGA/ICS terminology [10]. A gynecological examination was performed and, in case of prolapse, it was staged according to the POP-Q system.

Non-pregnant patients older than 18 years were included in the study if they had isolated SUI without surgical indication, confirmed with a standard 300 mL positive stress test. Exclusion criteria were a history of neoplasia, arrhythmia, congestive heart failure, recent deep venous thrombosis, fever, acute inflammatory diseases, or fractures in the area of treatment. Moreover, women with insufficient Italian language proficiency, a weight of more than 160 kg, neurostimulators, pacemakers, defibrillators, or ferromagnetic prostheses were excluded, as previously stated [J]. At the baseline (T0), the International Consultation on Incontinence Questionnaire-Short Form questionnaire (ICIQ-SF), the Female Sexual Function Index (FSFI-19) questionnaire, and the Incontinence Impact Questionnaire (IIQ-7) [35,36,37] were submitted and completed by all patients.

The ICIQ-SF questionnaire has been validated to measure the severity, frequency, and impact of urinary incontinence on quality of life [35]. The tool includes four questions, with the first three determining the total score: the leakage frequency, the perceived amount of leakage, and the level of impact on daily life [35]. The last item does not concur with the total score and is aimed to self-define the sub-type of incontinence [35]. This questionnaire showed high levels of validity, reliability, and sensitivity, and these parameters were evaluated through the use of standard psychometric tests [35]. The FSFI-19 questionnaire is a self-reported tool consisting of 19 items with a 5-point Likert scale addressing 6 domains of sexual function, including desire, lubrication, arousal, orgasm, pain, and satisfaction [36]. This instrument has consistently demonstrated satisfactory psychometric properties in evaluating the impact of several conditions on sexual well-being and the efficacy of different treatments [36]. Consequently, to investigate female sexual dysfunction at the baseline and after therapies, FSFI-19 represents one of the most valid, useful, popular, and powerful diagnostic tools [36]. For differentiating patients with and without sexual disorders, a cut-off of 26.5 points has been proposed to be the optimal [36]. The IIQ-7 questionnaire was introduced to investigate the impact of urinary incontinence on women’s daily life [37]. The questionnaire consists of seven items with the aim to evaluate the perceived feelings and impact of urinary incontinence on daily life and relationships [37]. Each item has four answers that participants use to individually self-evaluate the impact of urine leakage on daily activities in four domains: physical activity (items #1 and #2), travel (items #3 and #4), social activities (item #5), and emotional health (items #6 and #7) [37]. Based on psychometric tests, across different countries and cultures, this tool was associated with an excellent level of validity, acceptability, and reliability [37].

After proper counseling, patients underwent flat magnetic stimulation with Dr. Arnold (DEKA, Calenzano, Italy) according to the following protocol: eight sessions (twice a week) of 25 min each, using the “Weakness 1” protocol from sessions 1 to 4 and the “Weakness 2” protocol from sessions 5 to 8. The “Weakness 1” protocol involves a primary warm-up phase and muscle activation and a second phase of muscle work based on recovering tropism and muscle tone (20–30 Hz) in a trapezoidal shape. The “Weakness 2” protocol involves a warm-up and muscle activation phase followed by muscle work with the aim of increasing tropism (volume), and a muscle strength phase (40–50 Hz) in a trapezoidal shape.

At the end of the treatment (T1), a 300 mL stress test was required to assess the objective cure rate, and patients compiled again the ICIQ-SF, IIQ-7, and FSFI-19 questionnaires. The subjective cure rate was evaluated through the results from the Patient Global Impression of Improvement (PGI-I) questionnaire [38]. The PGI-I questionnaire is a 7-point scale that ensures the clinician can assess how much the patient’s disease has improved or worsened compared to a baseline state collected at the beginning of the treatment. This scale is described as follows: 1, very much improved; 2, much improved; 3, minimally improved; 4, no change; 5, minimally worse; 6, much worse; and 7, very much worse [38]. Subjective success was defined as an improvement in the PGI-I score (≤3). Three months after the end of the treatment (T2), the ICIQ-SF, IIQ-7, FSFI-19, and PGI-I questionnaires were resubmitted to the patients, and the stress test was repeated.

The local Ethics Committee approval was obtained (protocol code PF-MAGCHAIR). The scores obtained from the questionnaires were described as the median and interquartile range (IQR) after the failure of the normality check and were performed by using the Shapiro–Wilk test. Friedman’s non-parametric test [39] for repeated measures was then used to compare the IIQ-7, ICIQ-SF, and FSFI-19 questionnaire scores, as the small sample size did not allow obtaining normally distributed continuous variables, even after data transformation according to Blom’s method [40]. Durbin–Conover pairwise comparisons were used to check for significant differences between the three moments of data collection; this method was preferred over the classic Durbin test to maximize statistical power [41]. Prior to comparing the scores obtained throughout the study, we used the Mann–Whitney U test to check if relevant covariates such as body mass index, number of deliveries, and age produced any statistically significant differences in baseline scores. Confidence intervals for binomial proportions were calculated according to the methods suggested by Ross [42]. Significant differences between proportions were checked by using McNemar’s test, as the data came from repeated measures [43]. The significance threshold was established at 0.05 for all calculations; the analysis was conducted with R 4.1 (the R Core Team, Vienna, Austria, 2021) for MacOS^®^.

## 3. Results

Our study enrolled a total of 25 consecutive patients. Population characteristics are shown in Table 1. Baseline IIQ7 and ICIQ-SF scores were comparable by age, body mass index, and the number of deliveries (*p* > 0.05 for all calculations) as most women had normal weight (Me = 25.2, IQR = 3.10, eleven overweight and one obese with BMI = 31.8 kg/m^2^), and there was only one nulliparous in the sample. Baseline FSFI-19 scores showed a significant, albeit weak, correlation with age (rho = −0.411, *p* = 0.041) and BMI (rho = −0.473, *p* = 0.017). These two variables were, therefore, considered as covariates in the analyses regarding sexual function scores, while all other analyses were unadjusted. No women reported adverse effects during the treatment. Outcome measures of objective, subjective, and quality-of-life questionnaires at the baseline (T0), end of treatment (T1), and 3-month follow-up (T2) are summarized in Table 2. After the treatment, the decrease in the IIQ7 scores (bothersome level of leakages) was statistically significant compared to the baseline (*p* < 0.001), thus supporting the clinical usefulness of this treatment. However, at the three-months follow-up evaluation, the IIQ7 scores showed a statistically significant increase (*p* = 0.005), thus returning to levels comparable to the baseline condition of the patients (*p* = 0.135). Similarly, at the end of the treatment, we observed a statistically significant decrease in the ICIQ-SF scores compared to the baseline (*p* = 0.002). However, the ICIQ-SF values also increased significantly after three months from the end of the sessions, becoming comparable to bothersome baseline levels. Regarding sexual function, the differences observed in the conditions reported by the patients through the FSFI-19 questionnaire did not reach statistical significance, neither between the scores before treatment and at the end of the sessions nor between the latter and those obtained three months after the end of the rehabilitation program. Regarding the overall perception of improvement reported by the women, the PGI-I scores reported no statistically significant differences (*p* = 0.564) three months after the end of treatment (T2) compared to those obtained at the end of the sessions (T1) even after adjusting the analysis for overweight or obesity and the number of deliveries. With respect to objective outcomes, at the end of the rehabilitation program (T1), the number of women with negative stress tests was 10 out of 25 (40.0%). After three months (T2) 5 out of 25 (20.0%) patients maintained this result (proportion difference = −0.200, 95%CI = [−0.4225; 0.0533]), as shown in Table 2 and Table 3, and this decrease was statistically significant (*p* = 0.025).

## 4. Discussion

International guidelines recommend, as the first-line treatment for SUI, conservative management. Among all conservative treatment options including PFMT, functional electrical stimulation, and biofeedback, MS offers some advantages. Concerning PFMT, patients may not be able to recruit, contract, and adequately train the pelvic floor muscle thus reducing its effectiveness and consistency over time [44]. As a consequence, patients who underwent PFMT may show reduced compliance and adherence rates and notice a slow progression of the improvements [45]. Due to the use of endocavitary devices, both functional electrical stimulation and biofeedback can be badly tolerated or even not possible due to impaired vaginal habitability, such as in the case of lichen sclerosis, previous surgery, or radiation. Moreover, mild local discomfort and side effects may cause treatment discontinuation in up to 12% of patients [46]. On the contrary, MS is a type of passive rehabilitation with no adverse effects described, which does not involve endocavitary probes, and patients stay dressed. Moreover, unlike the electrical current, tissue impedance does not affect the conduction of electromagnetic energy. With all these aspects, MS can be defined as a non-invasive, standardizable, and safe conservative treatment option for urinary incontinence management. In particular, the latest innovation in magnetic stimulation technology is represented by flat magnetic stimulation technology. Flat magnetic stimulation induces strong muscle contractions through the induction of electrical currents in the context of pelvic floor neuromuscular tissue. This, consequently, induces more efficient muscle fiber hypertrophy and hyperplasia, changing the muscle’s structure. The hypertrophic effect of this technology on the skeletal muscles has been previously demonstrated. Leone et al. evaluated the impact of flat magnetic stimulation on the abdomens of 15 patients, demonstrating 1 month after the last treatment an increase in the abdominal muscle tissue thickness in the treated areas (lateral, upper, and lower abdomen) ranging from +14% to +23% [47]. Similarly, a significant (+15.4%) hypertrophy of the external urethral sphincter has been demonstrated in female patients with stress urinary incontinence [34]. However, the duration of this benefit and the optimal maintenance treatment schedule are still unknown.

For the first time, our study prospectively compared short- and medium-term outcomes of flat magnetic stimulation in patients with stress urinary incontinence. We found that, despite a certain persistence of the objective and subjective continence improvement, urinary-related quality-of-life tends to return to baseline values three months after treatment. Among the quality-of-life outcomes, a statistically significant reduction in the IIQ7 scores (a bothersome number of leakages) was observed after the treatment compared to the baseline but the IIQ7 scores significantly increased (*p* = 0.005), returning to levels comparable to the baseline condition at three months follow-up. Similarly, a statistically significant reduction in the ICIQ-SF scores at the end of the treatment compared to the baseline was followed by a significant increase in the ICIQ-SF values after three months from the end of the sessions, becoming comparable to the baseline. The subjective outcome evaluated by the PGI-I score showed no statistically significant differences (*p* = 0.564) three months after the end of treatment (T2) compared to the end of the sessions (T1), even after adjusting the analysis for overweight or obesity and the number of deliveries. In addition, after three months (T2), 5 out of 25 (20.0%) patients maintained a negative stress test compared to 10 out of 25 (40%) at the end of the rehabilitation program (T1). These findings indicate that a further eight-session cycle of treatment is probably indicated after 3 months since benefits are only partially still present at this time point.

To date, few pieces of evidence are available about the role of flat magnetic stimulation in the treatment of SUI, and there are none about the maintenance schedule. Lopopolo et al. evaluated improvement in the quality of life in 50 female patients with mixed urinary incontinence [48]. All patients underwent six sessions (twice a week) of 28 min each of Dr. ARNOLD magnetic stimulation. The first two minutes of warm-up were followed by the two protocols, Hypotonus/Weakness 1 and Hypotonus/Weakness 2. The ICIQ-UI-SF questionnaire, the Incontinence Questionnaire Overactive Bladder Module (ICIQ-OAB), and the IIQ-7 questionnaire were compiled at the baseline, during the treatment, and after three months. Quality of life improved from the second treatment session to the last one by 91%, 86%, and 98% according to the ICIQ-UI-SF, ICIQ-OAB, and IIQ-7 respectively. After three months, a small increase in scores was noticed, and the scores were better compared to the baseline; this can be probably explained by the return to a physiological hypotonus in the absence of long-term exercise.

Another study by Biondo et al. analyzed eighty-one female patients with urinary incontinence to evaluate the safety and the effectiveness of flat magnetic stimulation [49]. Women were divided into two groups: group A included 35 female patients who met the criteria for stress urinary incontinence, while group B enrolled 46 women with urge urinary incontinence. All patients underwent eight sessions of treatment for 28 min each twice a week for 4 straight weeks with the DR. ARNOLD system. Firstly, all patients started with a short warm-up phase followed by four sessions with the Hypotonus/Weakness 1 protocol and four sessions with the Hypotonus/Weakness 2 protocol for group A. There were eight sessions with the Overtone/Pain protocol (muscle work aimed at muscle inhibition) for group B.

Two questionnaires were completed before each treatment and at 3 months follow-up. The ICIQ-OAB questionnaire was compiled by the patients in group B, while the IIQ-7 questionnaire was assigned and filled out by the patients of group A. According to questionnaire results, there was an improvement in urge and stress urinary incontinence symptoms at the baseline and after treatment sessions at 3 months follow-up [49].

While specific data about the loss of hypertrophy on pelvic floor muscles due to detraining are not available, some studies have examined the effects of detraining on other muscles. In athletes’ hearts, the regression of the physiological left ventricular hypertrophy seems to occur already during the first month of detraining, with no further reduction between 1 and 3 months [50]. Regarding skeletal muscles, Narici et al. found a decrease of 4% in the muscle cross-sectional area (CSA) after a period of 40 days of detraining in the quadriceps muscles [51]. Similarly, Psillander et al. aimed to determine if a previously strength-trained leg would respond better to a period of strength training than a previously untrained leg, hypothesizing that the trained leg would have an enhanced hypertrophic response and an increased number of myonuclei compared with the untrained leg. Using muscle biopsies and ultrasounds, they showed that the increase in muscle thickness seen during the training period was completely lost after a 20-week period of detraining, but a relatively large increase in muscle thickness was observed during retraining in both the trained leg and the untrained leg (~10%) [52]. These findings are consistent with our study, which enlightens the necessity of performing retraining after a few months from the first stimulation to maintain the benefits in the long term. From the point of view of physiology, as reported by Terzoni et al., in a previous study on magnetic innervation, the lack of persistence of the results obtained with this rehabilitation method can be explained by the fact that, if no maintenance exercises are performed after the end of the stimulation program, muscular performance can rapidly decrease due to lack of exercise [53].

To date, this is the first study on women with stress urinary incontinence comparing short- and medium-term data about flat magnetic stimulation treatment to try to define an optimal maintenance schedule. Other strengths involve the prospective design and the multimodal evaluation of benefits, including objective cure rate, subjective impression of improvements, and multiple validated quality-of-life questionnaires. A limitation is the small sample size analyzed. Future research can include the evaluation of flat magnetic stimulation benefits in a larger population study compared to a control group. Another reasonable purpose would be to collect data after a prolonged period of observation, maybe after further sessions of treatment.

## 5. Conclusions

Our analysis concluded that flat magnetic stimulation represents a safe and effective stress urinary incontinence’s conservative treatment in terms of incontinence cure rate and quality-of-life improvement. However, despite a certain persistence of the objective and subjective continence improvement, the benefit in terms of quality of life tends to return to baseline values three months after the end of the treatment. These findings indicate that probably, after 3 months, a further cycle of treatment is indicated since benefits are only partially maintained after this timespan.

## Figures and Tables

**Table 1 healthcare-11-01730-t001:** Population characteristics and baseline (T0) findings. Continuous data as mean ± standard deviation. ICIQ-SF: International Consultation on Incontinence Questionnaire-Short Form questionnaire; FSFI-19: Female Sexual Function Index questionnaire; IIQ-7: Incontinence Impact Questionnaire.

Age (years)	60.9 ± 12.7
Parity (n)	1.9 ± 0.7
BMI (kg/m^2^)	25.4 ± 3.0
IIQ-7 score (T0)	33.7 ± 22.6
ICIQ-SF score (T0)	11.2 ± 3.6
FSFI-19 score (T0)	12.5 ± 11.2

**Table 2 healthcare-11-01730-t002:** Outcome measures of objective, subjective, and quality-of-life questionnaires at the baseline (T0), end of treatment (T1), and 3-month follow-up (T2). Data are reported as median and interquartile range except for stress test proportion expressed as absolute (relative) frequencies. ICIQ-SF: International Consultation on Incontinence Questionnaire-Short Form questionnaire; FSFI-19: Female Sexual Function Index questionnaire; IIQ-7: Incontinence Impact Questionnaire; PGI-I: Patient Global Impression of Improvement questionnaire.

Questionnaire	Baseline	End of Treatment	3-Month Follow-Up
IIQ-7	33.00 (38.50)	16.50 (11.00)	22.00 (22.00)
ICIQ-SF	12.00 (4.00)	8.00 (6.00)	10.00 (5.00)
FSFI-19	7.80 (22.80)	6.70 (22.30)	6.00 (22.20)
PGI-I	N/a	3.00 (2.00)	3.00 (2.00)
Positive stress test	25 (100%)	15 (60%)	20 (80%)

**Table 3 healthcare-11-01730-t003:** IIQ-7, ICIQ-SF, FSFI-19, and PGI-I scores and positive stress test rates comparisons among the endpoints of the study: the baseline (T0), end of treatment (T1), and 3-month follow-up (T2). *P*-values are provided. Durbin–Conover pairwise comparisons were performed to check for significant differences between the three moments of data collection. ICIQ-SF: International Consultation on Incontinence Questionnaire-Short Form questionnaire; FSFI-19: Female Sexual Function Index questionnaire; IIQ-7: Incontinence Impact Questionnaire; PGI-I: Patient Global Impression of Improvement questionnaire. N/A. not applicable. In bold statistically significant results.

	T0 vs. T1	T1 vs. T2	T0 vs. T2
IIQ-7	**<0.001**	**0.005**	0.135
ICIQ-SF	**0.002**	**0.034**	0.247
FSFI-19	0.394	0.495	0.864
PGI-I	N/A	0.564	N/A
Positive stress test	**0.001**	**0.025**	**0.025**

## Data Availability

The data presented in this study are available upon request from the corresponding author.

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
