# Peer review of "Flat Magnetic Stimulation for Stress Urinary Incontinence: A 3-Month Follow-Up Study"

_healthcare, 2023, doi:10.3390/healthcare11121730_

Round 1

Reviewer 1 Report

This research article examines the effectiveness of Flat Magnetic Stimulation (FMS) in treating urinary incontinence and sexual dysfunction in women. The study utilized three validated questionnaires - ICIQ-SF, FSFI-19, and IIQ-7 - to assess the severity, frequency, and impact of these conditions. The introduction is well-written and provides crucial information on the main topic. However, it is recommended that the authors include a paragraph discussing the impact of urinary incontinence on quality of life and the surgical options available to patients. The following papers may be useful for this purpose: 10.3390/jcm11216572 and 10.3390/jcm11195639.

The methods section is comprehensive and appears to be suitable for the study's objectives. Results are well-presented using tables. However, the sample size is small, and the absence of a randomized controlled trial makes it difficult to establish a causal relationship between the treatment and symptom improvement. This constitutes a significant limitation of the study.

The discussion is well-organized and based on the study's results. The conclusions align with the findings of the study. Overall, this research article provides valuable insights into the effectiveness of FMS in managing urinary incontinence and sexual dysfunction in women, although its limitations must be taken into account.

Author Response

Reviewer 1

This research article examines the effectiveness of Flat Magnetic Stimulation (FMS) in treating urinary incontinence and sexual dysfunction in women. The study utilized three validated questionnaires - ICIQ-SF, FSFI-19, and IIQ-7 - to assess the severity, frequency, and impact of these conditions. The introduction is well-written and provides crucial information on the main topic. However, it is recommended that the authors include a paragraph discussing the impact of urinary incontinence on quality of life and the surgical options available to patients. The following papers may be useful for this purpose: 10.3390/jcm11216572 and 10.3390/jcm11195639.

Added suggested ref in the text and enriched the introduction section

The methods section is comprehensive and appears to be suitable for the study's objectives. Results are well-presented using tables. However, the sample size is small, and the absence of a randomized controlled trial makes it difficult to establish a causal relationship between the treatment and symptom improvement. This constitutes a significant limitation of the study.

Added as limitation in the text

The discussion is well-organized and based on the study's results. The conclusions align with the findings of the study. Overall, this research article provides valuable insights into the effectiveness of FMS in managing urinary incontinence and sexual dysfunction in women, although its limitations must be taken into account.

Thank you.

Reviewer 2 Report

Dear authors, congratulations for your hard work. It is true, SUI represents a big portion of the female population, who seek conservative treatment and the options provided, especially PFMT, has limited success, providing it is adequately  performed.

Your proposal is interesting and seems promising if, and there is a big if, because you have scared data since your sample is small. Apart this major issue of your study, one more, in my humble opinion is the lack of control group. Imagine having two cohorts, with larger numbers of participants, comparing them. I am sure that you will agree.

Although your goal was to evaluate your method within short and medium tperiod of time, it would be beneficial if you had a fourth time mark, for example, 6 months after completing the treatment.

I surely agrre with you that perhaps you need another round of treatment, in order to evaluate wether it is beneficial and the results of it are permanent or more prominent at least.

Nevertheless, your approach has potentials and I storngly encourage you to collect more data, both in terms of bigger population, along with a control group, and in terms of prolonged period of observation.

I am looking forward to see your updated data.

Author Response

Reviewer 2

Dear authors, congratulations for your hard work. It is true, SUI represents a big portion of the female population, who seek conservative treatment and the options provided, especially PFMT, has limited success, providing it is adequately  performed.

Thank you.

Your proposal is interesting and seems promising if, and there is a big if, because you have scared data since your sample is small. Apart this major issue of your study, one more, in my humble opinion is the lack of control group. Imagine having two cohorts, with larger numbers of participants, comparing them. I am sure that you will agree.

Although your goal was to evaluate your method within short and medium tperiod of time, it would be beneficial if you had a fourth time mark, for example, 6 months after completing the treatment.

I surely agrre with you that perhaps you need another round of treatment, in order to evaluate wether it is beneficial and the results of it are permanent or more prominent at least.

Nevertheless, your approach has potentials and I storngly encourage you to collect more data, both in terms of bigger population, along with a control group, and in terms of prolonged period of observation.

I am looking forward to see your updated data.

Thank you for your suggestions. Added as limitation and added in the text as future research

Round 2

Reviewer 1 Report

The manuscript has been revised accordingly with the previous comments.

Reviewer 2 Report

Dear authrors, thank you for taking into account my suggestions. Best of luck.